# Exploring the Mechanism of Salvianolic Acid B against Myocardial Ischemia-Reperfusion Injury Based on Network Pharmacology

**DOI:** 10.3390/ph17030309

**Published:** 2024-02-28

**Authors:** Qianping Mao, Chongyu Shao, Huifen Zhou, Li Yu, Yida Bao, Yali Zhao, Jiehong Yang, Haitong Wan

**Affiliations:** 1School of Life Sciences, Zhejiang Chinese Medicine University, Hangzhou 310053, China; mqp971002@163.com (Q.M.); 18409441168@163.com (Y.Z.); 2School of Basic Medicine, Zhejiang Chinese Medicine University, Hangzhou 310053, China; paulscy86@hotmail.com (C.S.); zhouhuifen2320@126.com (H.Z.); yuli9119@126.com (L.Y.); yidabaojnu@foxmail.com (Y.B.); 3Research Institute of Traditional Chinese Medicine Encephalopathy, Zhejiang Chinese Medicine University, Hangzhou 310053, China

**Keywords:** MI/RI, Sal-B, network pharmacology, pharmacological mechanism, apoptosis

## Abstract

This study aimed to explore the mechanisms through which salvianolic acid B (Sal-B) exerts its effects during myocardial ischemia-reperfusion injury (MI/RI), aiming to demonstrate the potential pharmacological characteristics of Sal-B in the management of coronary heart disease. First, Sal-B-related targets and MI/RI-related genes were compiled from public databases. Subsequent functional enrichment analyses using the protein–protein interaction (PPI) network, gene ontology (GO), and the Kyoto Encyclopedia of Genes and Genomes (KEGG) predicted the core targets and approaches by which Sal-B counters MI/RI. Second, a Sal-B-treated MI/RI mouse model and oxygen–glucose deprivation/reoxygenation (OGD/R) H9C2 cell model were selected to verify the main targets of the network pharmacological prediction. An intersectional analysis between Sal-B and MI/RI targets identified 69 common targets, with a PPI network analysis highlighting caspase-3, c-Jun N-terminal kinase (JNK), and p38 mitogen-activated protein kinase (p38) as central targets. GO and KEGG enrichment analyses indicated remarkable enrichment of the apoptosis pathway among these targets, suggesting their utility in experimental studies in vivo. Experimental results demonstrated that Sal-B treatment not only mitigated myocardial infarction size following MI/RI injury in mice but also modulated the expression of key apoptotic regulators, including Bcl-2-Associated X (Bax), caspase-3, JNK, and p38, alongside enhancing the B-cell lymphoma-2 (Bcl-2) expression, thereby inhibiting myocardial tissue apoptosis. This study leveraged an integrative network pharmacology approach to predict Sal-B’s potential targets in MI/RI treatment and verified the involvement of key target proteins within the predicted signaling pathways through both in vivo and in vitro experiments, offering a comprehensive insight into Sal-B’s pharmacological mechanism in MI/RI management.

## 1. Introduction

In China, over 330 million people suffer from cardiovascular disease, and the mortality rate from coronary heart disease exceeded 0.1% in 2019 [1]. Acute myocardial infarction is a clinical manifestation of coronary heart disease that can cause heart failure and is the main cause of cardiovascular disease-related deaths [2]. Timely blood flow recovery can reduce myocardial injury and has become the main treatment for myocardial infarction [3,4]. However, reperfusion itself can lead to additional myocardial injury, known as myocardial ischemia-reperfusion injury [5]. Currently, the discussion about this pathological process is focused on oxidative stress, calcium overload, inflammatory reactions, and energy metabolism disorders, and the process has been verified in various animal models [6,7]. However, verapamil as a positive drug for MI/RI may increase reactive oxygen species (ROS) levels, promote oxidative stress, and cause heart failure [8]. Some studies have shown that oxidative stress is the main pathological process in MI/RI [9]. When MI/RI damage occurs, the mitochondrial membrane potential changes, permeability increases, and a large amount of ROS is released, causing cell senescence and apoptosis [10].

ROS are the result of regular cell metabolism and are in relative equilibrium under normal conditions [11]. During reperfusion, a large amount of ROS is released, breaking the equilibrium, and the activity of the oxidant scavenging enzyme system (superoxide dismutase, glutathione) decreases [11], mediating membrane lipid peroxidation and MDA production [12], causing mitogen-activated protein kinase signaling pathway activation [13], JNK and P38 phosphorylation [14,15], and cell apoptosis [16,17].

Salvianolic acid B is the most abundant water-soluble compound derived from *Salvia miltiorrhiza* root and is a natural bioactive antioxidant [18]. It has multifunctional effects on oxidative stress, inflammation, aging, and human cancer [19,20,21,22,23,24,25], effectively inhibiting ROS production and reducing the production of lipid peroxidation products such as MDA, thus inhibiting apoptosis [26]. In the astrocyte and microglia OGD/R injury model [27], salvianolic acid B was observed to counteract the internalization of connexin 43 (Cx43) within the plasma membrane prompted by OGD/R-induced damage. However, it did not significantly alter the overall cellular levels of Cx43. This action facilitated and enhanced intercellular communication, diminished the activation of hemichannels, and conferred protection against OGD/R-induced cellular damage. In a renal ischemia-reperfusion injury model, salvianolic acid B inhibited the NOD-like receptor thermal protein domain-associated protein 3 (NLRP3) activation by directly activating the nuclear expression of nuclear factor erythroid 2-related factor 2 (Nrf2), thus inhibiting focal death in MI/RI [28]. Salvianolic acid B has been widely used in the cardiovascular field [29]. Research indicates that Sal-B possesses therapeutic and protective properties against cardio-cerebrovascular diseases [30,31]. Some studies have shown that Sal-B alleviates the injury associated with diabetic cardiomyopathy by inhibiting IGFBP3 expression [18]. It also alleviates myocardial infarction injuries by inhibiting MEG3 [32]. Despite its potential efficacy in treating MI/RI, the precise mechanisms underlying its action remain to be fully elucidated.

To explore the anti-MI/RI effect of Sal-B through network pharmacological research and experimental verification, we used cyberpharmacology. Cyberpharmacology is a computer technology for new drug discovery founded by Hopkins in 2007 [33]. It is an interactive network based on drug-target diseases, including chemical informatics, bioinformatics, cyberbiology, and pharmacology [34,35]. It is a comprehensive tool that can systematically reveal the complex network relationship between multiple components and potential mechanisms of traditional Chinese medicine formulations from a systematic point of view [36]. Several databases have been used to predict Sal-B targets. The possible mechanism of the anti-MI/RI effect of Sal-B was discussed using graphene oxide biological process analysis and KEGG pathway enrichment studies.

In this study, we combined in vivo and in vitro models to explore the mechanisms of Sal-B during MI/RI and demonstrate the potential pharmacological characteristics of Sal-B in the clinical treatment of coronary heart disease.

## 2. Results

### 2.1. Network Pharmacology Analysis of Potential Targets of Sal-B in the Treatment of MI/RI

Through screening databases and Swiss target prediction, 198 Sal-B targets were identified. To obtain targets related to MI/RI, we screened DisGeNET (https://www.disgenet.org/ 8 July 2021) and gene cards with “MI/RI” as the keyword and obtained 1264 targets. The 1264 targets related to MI/RI and the 292 targets of Sal-B were combined using a Venn diagram. There were 69 potential MI/RI targets between known Sal-B-related and compound targets (Figure 1B).

To assess the priority of common targets, the 69 MI/RI-related Sal-B potential targets were imported into the STRING11.0 database (https://string-db.org/ 8 July 2021) for network topology analysis (a combined score of > 0.4 was the criterion). These targets were analyzed using Cytoscape software (Figure 1C). The top ten targets were ALB, CASP3, ANXA5, NOS1, SRC, NOS3, MAPK14, MAPK8, MAPK1, and PPARG.

For further analysis, the Database for Annotation, Visualization, and Integrated Discovery database was used to analyze the graphene oxide enrichment of the 69 targets in this study (Figure 1D). The gene ontology enrichment analysis results showed that the apoptosis signaling pathway ranked second.

The enrichment of KEGG signaling pathways demonstrated that the Sal-B treatment of myocardial ischemia-reperfusion injury mainly involved the following: cell death-related signaling (such as apoptosis), inflammation reaction-related signaling (such as lipid and atherosclerosis and MAPK pathways), and oxidative stress reaction-related signaling pathways (such as ROS and MAPK pathways).

### 2.2. Salvianolic Acid B Attenuates Myocardial Injury Induced by Myocardial Ischemia-Reperfusion in Mice

We evaluated the therapeutic effect of Sal-B on MI/RI in vivo (Figure 2A). We carried out a model experiment of myocardial ischemia-reperfusion in mice and found no significant difference in the area at risk (AAR) to left ventricle (LV) ratio between the normal and model groups and the Sal-B groups (Figure 2B). Compared to the MI/RI group, the Sal-B group showed a significantly decreased infarct size (INF) in the AAR ratio (Figure 2C).

According to hematoxylin and eosin staining (HE) staining, the myocardial morphology of normal mice was normal, the nucleus was purplish-red, and the cytoplasm was light red. In the model group, the infiltration of inflammatory cells was severe, pyknosis and necrosis of the nucleus were evident, and the cell boundary was vague. In the salvianolic acid B group, inflammatory cell infiltration was reduced, nuclear pyknosis and necrosis were improved, and the therapeutic effect of 20 mg/kg/day Sal-B was enhanced (Figure 2D).

### 2.3. Salvianolic Acid B Can Reduce Cell Apoptosis after OGD/R

The cell counting kit-8 assay (CCK-8) results showed that under normoxic conditions, there was no significant change in cell activity after treatment with Sal-B (0–20 μM) (Figure 3A). Subsequently, 1, 5, and 10 μM were selected as administration doses. Compared to the OGD/R group, the Sal-B group showed significantly increased cell viability (Figure 3B).

The terminal deoxynucleotidyl transferase-mediated dUTP-biotin nick end labeling assay (TUNEL) was used to detect apoptosis after OGD/R. The apoptosis rate in the Sal-B group was significantly lower than that in the OGD/R group (Figure 3C,D).

### 2.4. Effect of Salvianolic Acid B on Apoptotic Proteins

B-cell lymphoma-2 (Bcl-2), Bax, and cleaved caspase-3 play important roles in the apoptosis of intersecting cells. Compared with the normal group, the ratio of Bcl-2/Bax in the OGD/R and MI/RI model groups decreased significantly, while the ratio of cleaved caspase-3/caspase-3 significantly increased. Sal-B treatment ameliorated these changes (Figure 4A–F). These results suggest that Sal-B can significantly reduce the apoptosis induced by myocardial ischemia-reperfusion injury.

### 2.5. Salvianolic Acid B Scavenges Oxygen-Free Radicals and Protects Mitochondrial Function

Oxidative stress plays an important role in both mitochondrial function and apoptosis. When ROS are released too much, they will aggravate the damage of oxidative stress. In this study, compared with the normal group, reactive oxygen species were significantly increased in the OGD/R group, and salvianolic acid B treatment remarkably downregulated OGD/R-induced ROS release (Figure 5A,B).

Lipid peroxidation produces malondialdehyde (MDA), which causes cytotoxicity, and compared with the normal group, the malondialdehyde of the MI/RI group was significantly increased, and the salvianolic acid B treatment reversed this change (Figure 5C).

Mitochondrial membrane potential represents mitochondrial dysfunction and cell damage. Mitochondrial membrane potential was measured to evaluate the protective effect of Sal-B against mitochondrial membrane injury. Compared to the normal group, the ratio of the Red/Green method in the model group decreased significantly, and the Sal-B treatment restored this change (Figure 5D,E).

### 2.6. The Protective Effect of Sal-B on Cells May Be Associated with Sirt 1/MAPK

Silent information regulator 1 (SIRT1) and MAPK-related proteins JNK and P38 are involved in cardiomyocyte apoptosis after hypoxia/reoxygenation [37]. Therefore, we tested the expressions of these proteins using immunoblotting. Compared with the normal group, silent information regulator 1 expression in the model group was significantly decreased, and downstream JNK and P38 phosphorylation were downregulated, whereas salvianolic acid B treatment increased the expression of SIRT1 and weakened the phosphorylation of JNK and P38, thus protecting the cells (Figure 6A–H).

## 3. Discussion

Salvianolic acid B has strong antioxidant and antiapoptotic abilities [38]. In this paper, based on the network pharmacological analysis combined with experimental verification, the mechanism of Sal-B against myocardial ischemia-reperfusion injury is discussed. The related targets (p38, JNK, and caspase-3) and related pathways (MAPK signaling pathway) were identified via network analysis. The MI/RI model of C57BL/6 mice and the OGD/R model of H9C2 cells further verified that Sa-B inhibits MI/RI-induced apoptosis. This study provides inaugural evidence that Sal-B can alleviate ROS release and inhibit the activation of the MAPK signaling pathway by restoring the SIRT1 expression, thereby affording cardioprotection against MI/RI.

During reperfusion, a large amount of ROS is released. Excessive ROS production leads to mitochondrial membrane damage and membrane potential changes, resulting in a mitochondria-dependent apoptosis pathway [3], causing apoptosis [39]. In this process, the dynamic balance between Bax (proapoptotic protein) and Bcl-2 (antiapoptotic protein) is disrupted [40], causing mitochondrial depolarization [41], the disruption of membrane integrity [42], and the release of proapoptotic factors (cytochrome C) into the cytoplasm [43]. Caspase cascade activation leads to caspase-3 activation in the executor [44,45]. Lytic caspase-3 is the active form of caspase-3 [46], which is the main cleavage enzyme that promotes apoptosis [47]. Some studies have shown that Sal-B can alleviate diabetic endothelial dysfunction by downregulating endothelial cell apoptosis [48]. In our study, salvianolic acid B attenuated the change in the early membrane potential of mitochondria, increased the ratio of Bcl-2/Bax, and inhibited lytic caspase-3 activation, thus increasing the survival rate of H9C2 cells after OGD/R. Moreover, salvianolic acid B decreased ROS release in H9C2 cells induced by OGD/R and reduced MDA production. Furthermore, the CCK-8 and TUNEL assay results show that Sal-B reduced the infarct area and degree of tissue injury in the myocardial tissue of C57BL/6 mice after MI/RI, indicating its antiapoptotic and oxidative stress functions.

The main members of the MAPK signaling pathway are considered proximal effectors of mitochondrial-dependent apoptosis. In particular, p38-MAPK and JNK-MAPK play key roles in apoptosis signaling [15,49]. When the MAPK pathway is activated, it increases the Bax expression and caspase-3 activation and inhibits Bcl-2 expression, which leads to apoptosis [50,51,52]. SIRT1 is a member of the deacetylase family of sirtuins (SIRTs) and is involved in the regulation of aging, oxidative stress, and apoptosis [53,54]. Some studies have shown that LF10 inhibits MAPK phosphorylation and attenuates the injury caused through diabetic cardiomyopathy by restoring SIRT1 [55]. Resveratrol (a SIRT1 agonist) inhibits the phosphorylation of JNK and p38 by activating SIRT1, deacetylating AKT, and attenuating skin damage [56]. In a model of MI/RI, SIRT1 activation also inhibits MAPK phosphorylation to reduce apoptosis [37]. Our results show that H9C2 cells pretreated with Sa-B restored myocardial SIRT1, inhibited JNK and p38 phosphorylation in the MAPK pathway and apoptosis, and alleviated myocardial injury.

In summary, Sal-B may increase SIRT1 activity, inhibit the phosphorylation of JNK and p38, reduce ROS release, and inhibit apoptosis through the SIRT1/MAPK pathway. This study enhances our comprehension of Sal-B’s mechanisms of action and contributes further insights into its potential as a therapeutic agent for myocardial ischemia-reperfusion (I/R) injury. Because of experimental limitations, we chose only the SIRT1/MAPK pathway and apoptosis mechanism to verify the effects of Sal-B in the treatment of myocardial ischemia-reperfusion, ignoring other pathways and targets. Other important targets and pathways will be further verified in the future, including acetylated-SOD2 (Ac-SOD2) in SOD2 to eliminate ROS and protect the myocardium [57]. NAD (P) H quinone dehydrogenase 1 (NQO1) exerts a protective effect on various metabolic oxidative stress responses after induction [58].

## 4. Materials and Methods

### 4.1. Material

Salvianolic acid B (CAS Number: 121521-90-2) was obtained from Chengdu Efa Biotechnology (Chengdu, China). The cell counting kit-8 (CCK-8, CAS Number:BS350B), mitochondrial membrane potential assay kit with JC-1 (CAS Number: C2006), ROS assay kit (CAS Number: S0033S), One Step TUNEL Apoptosis Assay Kit (CAS Number: C1089), DAPI Staining Solution (CAS Number: C1006), Lipid Peroxidation MDA Assay Kit (MDA, CAS Number: S0131S), and glyceraldehyde-3-phosphate dehydrogenase (GAPDH) antibody (CAS Number: AF5009) were purchased from Beyotime Biotechnology (Shanghai, China). Additionally, 2, 3, 5-triphenyltetrazolium chloride (CAS number: T8877-5G) was purchased from Sigma-Aldrich (Saint Louis, MO, USA). We purchased 0.5% Evans Blue dye (CAS Number: G1810) from Solarbio (Beijing, China). Bax (CAS Number: ab32503), Bcl-2 (CAS Number: ab196495), caspase-3 antibody (CAS Number: ab184787), and tubulin (CAS Number: ab176560) antibodies were obtained from Abcam (Cambridge, MA, USA). JNK antibody (CAS Number: 9252T), P38 antibody (CAS Number: 41666S), and SIRT1 antibody (CAS Number: 2493S) were purchased from Cell Signaling Technology (Danvers, MA, USA). The p-JNK antibody (CAS Number: 80024-1-RR) and P-P38 antibody (CAS Number: 28796-1-AP) were purchased from Proteintech (Danvers, MA, USA). The cleaved caspase-3 antibody (CAS Number: AF7022) was purchased from Affbiotech (Danvers, MA, USA).

### 4.2. Establishment of the Candidate Target Database

PharmMapper (http://lilab-ecust.cn/pharmmapper/ 8 July 2021, Version 2017) and Swiss Target Prediction (https://www.swisstargetprediction.ch/ 8 July 2021) databases were used to screen the target genes of the selected candidates in Section 2.1. Both web servers are online tools for predicting potential drug targets [59].

The target genes involved in the process of MI/RI were collected from two databases: Gene Cards (https://www.genecards.org/ 8 July 2021) and DisGeNET [60] (http://www.disgenet.org/ 8 July 2021). The candidate genes were collected according to the following parameters: a score gad > 0.1 in DisGeNET and a relevance score > 1 in Gene Card.

Finally, the Sal-B-targets and MI/RI-target databases were uploaded to the BioinfoGP (https://bioinfogp.cnb.csic.es/tools/venny/ 8 July 2021) platform to select potential therapeutic targets for the following analysis.

### 4.3. Construction of the PPI Network

The protein–protein interaction data were obtained by entering the common Sal-B and MI/RI targets into the String database. Subsequently, Cytoscape 3.8.0 software was used to construct the PPI networks based on the degree values.

### 4.4. Bioinformatic Annotation

In the enrichment section of the analysis page, key targets were uploaded to the GO enrichment analysis and KEGG pathways, and then the possible pathways of Sal-B were screened.

### 4.5. Animal Model

Male C57BL/6 mice (20 g–25 g) were purchased from Zhejiang Chinese Medical University and were fed in an SPF animal room for 1 week to acclimatize (free access to food and water, temperature 23 ± 1 °C, humidity 60–70%, 12 h light/dark cycle).

The animal research program was approved by the Animal Care and Use Committee of the Zhejiang Experimental Animal Research Center of Traditional Chinese Medicine. [License No. SYXK (Zhe) 2021–0012].

For MIRI induction, 32 male C57BL/6 mice were anesthetized with 0.3% pentobarbital sodium (50 mg/kg) and connected to an animal ventilator. Then, a 1.5-cm incision was made between the third and fourth ribs to expose the left anterior descending artery. The left anterior descending coronary artery was encircled with a 7-0 nylon suture.

Reperfusion was performed after myocardial ischemia for 45 min, followed by reperfusion for another 24 h. Low-dose Sal-B treatment (10 mg/kg), high-dose Sal-B treatment (20 mg/kg), or saline alone (MI/RI) was administered 7 d before the surgical procedure and once after the operation.

### 4.6. Staining with 2,3,5-Triphenyltetrazolium Chloride

Myocardial infarct areas were determined using Evans blue-TTC double staining. After the MI/RI procedure, 1% Evans blue solution was injected retrogradely into the aorta to mark the at-risk area. The hearts were excised and frozen at −80 °C for 2 h and then cut into slices, followed by incubation in 1% 2, 3, 5-triphenyltetrazolium chloride solution in the dark for 15 min at 37 °C to identify the infarcted myocardium after overnight incubation with 4% paraformaldehyde. Infarct size was calculated as the percentage of infarcted area to the abnormal area × 100%.

### 4.7. Hematoxylin and Eosin Staining

After the MI/RI procedure, the heart was removed and fixed at 4 °C in 4% paraformaldehyde for 24 h. The heart was then embedded in paraffin, sectioned at 4 μm intervals, and sectioned for HE staining. The sections were observed under a microscope and photographed.

### 4.8. Cell Culture

Dulbecco’s modified eagle medium (DMEM) with 10% fetal bovine serum was added to the dish so that the H9C2 cells were there, and then 100 μg/mL streptomycin was added in a humidified incubator at a constant temperature of 37  °C and 5% CO_2_.

### 4.9. Oxygen–Glucose Deprivation/Reoxygenation Model

When H9c2 cells reached 50% confluence, the medium was discarded, and the cells were washed thrice with phosphate-buffered saline (PBS). Different Sal-B concentrations (0, 1, 5, and 10 µM) were then added to the cell-containing medium and cultured for 24 h. We then discarded the medium, added serum-free and sugar-free medium to the H9C2 cells, and cultured them for 4 h in a cell incubator in an atmosphere of 1% O_2_, 94% N_2_, and 5% CO_2_ at 37 ℃. After 4 h, the serum-free and sugar-free media were replaced with a fresh complete medium, and cells were placed in an atmosphere of 5% CO_2_ at 37 °C for another 24 h of culture.

### 4.10. Cell Viability Was Determined Using CCK 8

The cells (0.6 × 10^4^ cells/well in 100 μL medium) were cultured in 96-well plates. At the end of the ODG/R procedure, cell viability was measured using the CCK-8 assay, and 10 μL of CCK-8 was added to each well for 2 h. Absorbance was measured at 450 nm using a microplate reader. The results were expressed as a percentage of the optical density (OD) at 450 nm measured in the control cells.

### 4.11. Assessment of Oxidative Stress

After the MI/RI procedure, the MDA levels in heart tissues were determined using MDA kits per the manufacturer’s protocols (Beyotime Biotechnology).

### 4.12. Reactive Oxygen Species Detection

At the end of the ODG/R procedure, the H9C2 cells were incubated with 10 µM 2′,7′-dichlorodihydrofluorescein diacetate solution in serum-free cell culture medium in the dark for 20 min at 37 °C and then analyzed via flow cytometry.

### 4.13. Mitochondrial Membrane Potential

Changes in the mitochondrial membrane potential were detected at the end of the ODG/R procedure using the JC-1 kit (CAS Number: C2006).

### 4.14. Cell Apoptosis

At the end of the OGD/R treatment, the apoptosis of H9C2 cells was examined using a TUNEL assay, and the nuclei were stained with DAPI: apoptosis rate (%) = the number of TUNEL-positive and DAPI-positive cells × 100%

### 4.15. Western Blotting Analysis

The cardiomyocytes were seeded into six-well plates, and at the end of the OGD/R treatment, the total protein from H9c2 cells was lysed using RIPA lysis buffer, and the total protein was measured using a bicinchoninic acid assay (BCA) kit.

After the MI/RI procedure, the tips of the mouse hearts were ground with liquid nitrogen and then added to RIPA lysis buffer containing protease inhibitors to collect the total protein. The total protein concentration was measured using a BCA assay kit.

Protein samples (25 μg) were loaded on sodium dodecyl sulfate–polyacrylamide gel electrophoresis (SDS-PAGE) gels via electrophoresis and transferred to a polyvinylidene difluoride (PVDF) membrane, which was blocked with tris-buffered saline with tween 20 (TBST) buffer containing 5% skim milk for 1 h and then incubated with the following primary antibodies at 4 °C overnight: anti-Bcl-2 (1:2000), anti-Bax (1:2000), anti-cleaved caspase3 (1:2000), anti-phosphorylated- (p-) p38 (1:2000), anti-p-JNK (1:2000), anti-total p38 (1:2000), anti-total JNK (1:2000), and anti-GAPDH (1:2000).

The secondary antibodies (1:8000) were diluted in TBST buffer. The bands were developed using an enhanced chemiluminescence (ECL) detection reagent. The protein band intensities were collected using the Azure C300 chemiluminescence imaging system and measured using ImageJ1.8.0 software.

## 5. Conclusions

In conclusion, this study demonstrated that Sal-B attenuated myocardial ischemia-reperfusion injury, alleviated hypoxic reoxygenation injury in H9C2 cells, and suppressed ROS release. Furthermore, Sal-B attenuated the changes in mitochondrial membrane potential in H9C2 cells. Therefore, Sal-B has great potential for the treatment of MI/RI.

## Figures and Tables

**Figure 1 pharmaceuticals-17-00309-f001:**
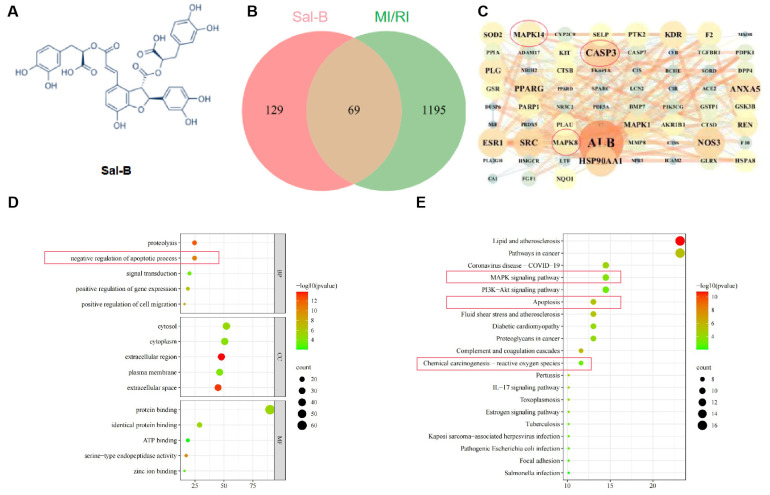
Network pharmacology analysis of potential targets of Sal-B in the treatment of MI/RI. (**A**) Salvianolic acid B structural formula. (**B**) Venn diagram illustrating the cross-targets of Sal-B and MI/RI. (**C**) A protein–protein interaction network depicting potential Sal-B targets for MI/RI intervention. (**D**) Top five categories in gene ontology enrichment analysis (BP represents the biological progress of core targets, CC represents the cellular components of the core target, and MF represents the molecular function of the core target). (**E**) The first 20 signaling pathways identified through the Kyoto Encyclopedia of Genes and Genomes (KEGG) analysis.

**Figure 2 pharmaceuticals-17-00309-f002:**
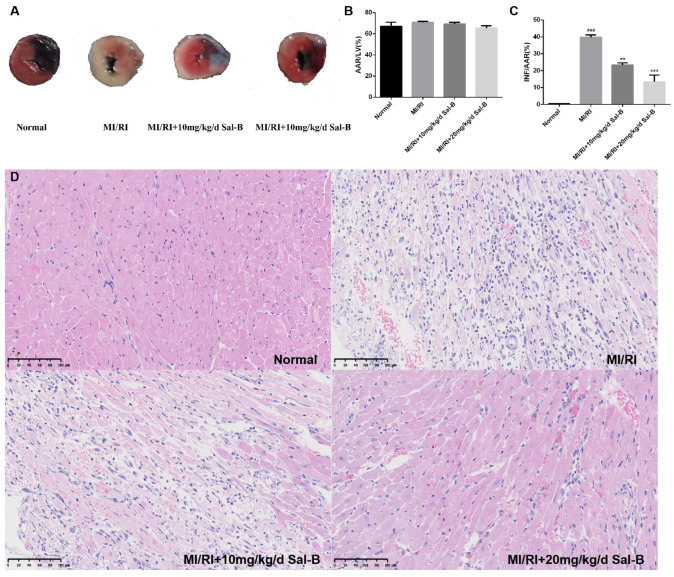
Salvianolic acid B attenuates myocardial injury induced by myocardial ischemia-reperfusion injury (MI/IR) in mice. (**A**) After MI/RI modeling, the mouse heart was stained with Evans blue/2, 3, 5-triphenyltetrazolium chloride (TTC) double staining to delineate the infarcted area (white), area at risk (red), and normal area (blue). (**B**) Analysis using 2, 3, 5-Triphenyltetrazolium chloride revealed the area at risk (AAR) to left ventricle (LV) ratio, where AAR constitutes the combined infarct and risk areas, and LV denotes the unaffected myocardial region. (**C**) Results from 2, 3, 5-Triphenyltetrazolium chloride staining indicated the infarct size (INF) relative to the area at risk (AAR), with INF representing the infarcted region and AAR encompassing both the infarct and risk areas. (**D**) Hematoxylin and eosin staining of the myocardium. Sample size: n = 3. Statistical significance is indicated as follows: compared with the normal group, ^###^
*p* < 0.001; compared with the model group, ** *p* < 0.01, *** *p* < 0.001.

**Figure 3 pharmaceuticals-17-00309-f003:**
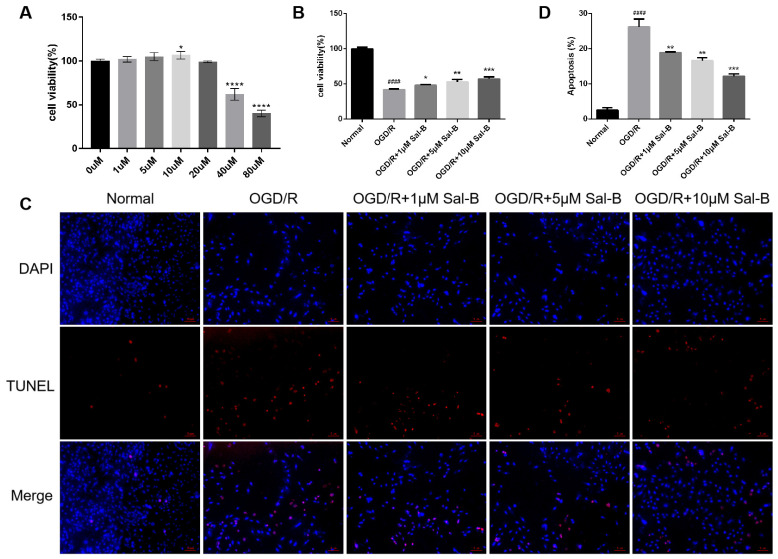
Salvianolic acid B can reduce cell apoptosis after OGD/R. (**A**) The effect of Sal-B (0 μM–80 μM) on H9C2 cell viability under normoxic conditions assessed using the CCK-8 assay, with results measured according to optical density (OD). (**B**) Cell viability evaluated according to optical density (OD) using the CCK-8 kit. (**C**) Apoptotic cells identified using TUNEL staining (red), with nuclei counterstained using 2-(4-Amidinophenyl)-6-indolecarbamidine dihydrochloride (DAPI) (blue) (scale bar = 100 μM). (**D**) Apoptosis quantified as the percentage of total cells. Sample size: n = 3. Statistical significance is denoted as follows: compared with the normal group, ^####^
*p* < 0.0001; compared with the model group, * *p* < 0.05, ** *p* < 0.01, *** *p* < 0.001, **** *p* < 0.001.

**Figure 4 pharmaceuticals-17-00309-f004:**
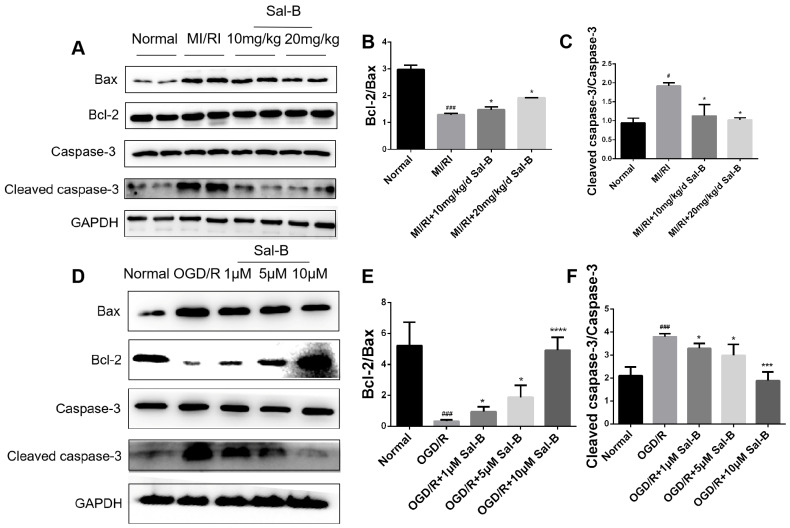
Effect of salvianolic acid B on apoptotic proteins. (**A**–**F**) Western blotting was used to evaluate the Bcl-2, Bax, cleaved caspase-3, and caspase-3 expression levels across the normal, model, and Sal-B groups, both in vivo and in vitro. Tubulin levels were also evaluated to confirm equal loading. Sample size: n = 3. Statistical significance is indicated as follows: compared with the normal group, ^#^
*p* < 0.05, ^###^
*p* < 0.001; compared with the model group, * *p* < 0.05, *** *p* < 0.001, **** *p* < 0.0001.

**Figure 5 pharmaceuticals-17-00309-f005:**
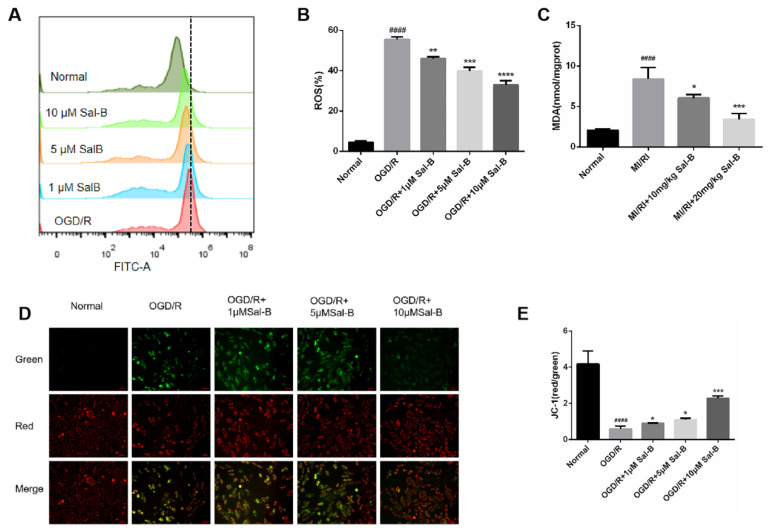
Salvianolic acid B scavenges oxygen-free radicals and protects mitochondrial function. (**A**,**B**) Evaluation of ROS content through ROS assay. (**C**) Evaluation of MDA content using the MDA method. (**D**,**E**) The value of ΔΨM evaluated based on 5,5′,6,6′-Tetrachloro-1,1′,3,3′-tetraethyl-imidacarbocyanine iodide 5,5′,6,6′-Tetrachloro-1,1′,3,3′-tetraethyl-imidacarbocyanine iodide (JC-1) staining (scale bar = 50 μm), the mitochondrial high membrane potential in red (normal cells) and mitochondrial low membrane potential in green (damaged cells). Sample size: n = 3. Statistical significance is indicated as follows: compared with the normal group, ^####^
*p* < 0.0001; compared with the model group, * *p* < 0.05, ** *p* < 0.01, *** *p* < 0.001, **** *p* < 0.0001.

**Figure 6 pharmaceuticals-17-00309-f006:**
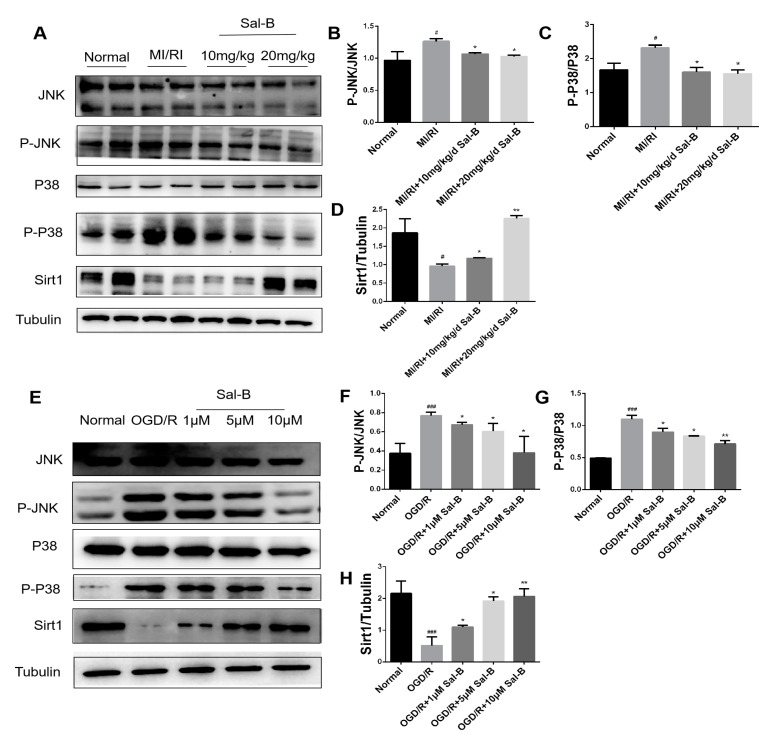
Salvianolic acid B against MI/RI via the SIRT 1/MAPK pathway. (**A**–**H**) Western blotting was used to evaluate the expressions of SIRT1, P-JNK/JNK, and P-P38/P38 in the normal, model, and salvianolic acid B groups both in vivo and in vitro. The tubulin levels were also evaluated to confirm equality. Sample size: n = 3. Statistical significance is indicated as follows: compared with the normal group, ^#^
*p* < 0.05, ^###^
*p* < 0.001; compared with the model group, * *p* < 0.05, ** *p* < 0.01.

## Data Availability

The data presented in this study are available from the corresponding author upon request.

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
