# Peer review of "Exploring the Mechanism of Salvianolic Acid B against Myocardial Ischemia-Reperfusion Injury Based on Network Pharmacology"

_pharmaceuticals, 2024, doi:10.3390/ph17030309_

Round 1

Reviewer 1 Report

Comments and Suggestions for Authors

Last year, Li N, (PMID: 37680709) reviewed that Network pharmacology-based analysis of potential mechanisms of myocardial ischemia-reperfusion injury by total salvianolic acid injection. This review shows total hypothetical points of the MS and Jing Z, (PMID: 27557491) reported Salvianolic acid B, is a novel autophagy inducer that indicates the cell signaling pathway of autophagy induced MI-RI.

So I could not find any scientific merits of your MS.

Author Response

Dear editor and Reviewer,

Thank you for your careful reading and thoughtful comments on our previous manuscript. Those comments are very valuable and helpful to revise and improve our manuscript for us.

Response: Li N, (PMID: 37680709) reviewed that Potential mechanisms of TSI in preventing myocardial ischemia and reperfusion in PI3K signaling, JAK-STAT signaling, Calcium signaling, Nuclear receptor signaling, HIF-1 signaling, Cell Cycle, and Apoptosis, while TSI is mainly composed of 14 compounds. Jing Z, (PMID: 27557491) reported that salvianolic acid B exerts antitumor effects by inducing autophagy and inhibiting the AKT / mTOR pathway.

In this paper, we mainly directed salvianolic acid B, identified MAPK pathway through network pharmacological screening, and played antioxidant and anti-apoptosis with Sirt 1 mediated MAPK pathway. In a model of MI/RI, SIRT1 activation also inhibits MAPK phosphorylation to reduce apoptosis. Our results showed that H9C2 cells pretreated with Sa-B had restored myocardial SIRT1, inhibited JNK and p38 phosphorylation in the MAPK pathway and apoptosis, and alleviated myocardial injury. In order to provide a better understanding of the working principle of Dan phenol acid B and its potential prevention mechanism in myocardial ischemia and reperfusion injury.

Reviewer 2 Report

Comments and Suggestions for Authors

The submitted manuscript detailing the investigation led by Qianping Mao et al. into the therapeutic potential of Salvianolic Acid B (Sal-B) in addressing myocardial ischemia-reperfusion injury (MI/RI) for potential application in coronary heart disease treatment is both compelling and noteworthy. Employing a network pharmacology approach, the research successfully identifies 69 key targets associated with Sal-B and MI/RI, highlighting pivotal targets such as caspase-3, JNK, and p38. Through rigorous experimental validation conducted in mice and cells, the study substantiates Sal-B's efficacy in diminishing infarction size and suppressing the expression of apoptosis-related proteins.

This research offers a comprehensive insight into the pharmacological mechanisms underlying Sal-B's efficacy in MI/RI treatment, seamlessly integrating computational predictions with robust experimental data. Consequently, I find the presented work to be of substantial merit, and I recommend its publication in its current form with only minor modifications.

The following minor comments are provided for your consideration:

1.     Abbreviations: Ensure that all abbreviations are defined upon their initial mention in the manuscript. Additionally, make sure that abbreviations in the figure legends are also clearly defined.

2.     Figure 2: The images in Figure 2 exhibit poor quality. I recommend replacing them with higher resolution and larger images to enhance clarity.

3.     Figure 3: The presentation of data in Figure 3 appears congested, making it challenging to interpret. I propose dividing Figure 3 into two separate figures to improve readability and comprehension.

Addressing these minor comments will contribute to the overall refinement of the manuscript.

Author Response

Dear reviewer: Thank you for your careful reading and thoughtful comments on our previous manuscript. These comments are very valuable and help to revise and improve our manuscript. We modified and polished the format and writing issues of the article, and marked them in red in the word document. Modified Figure 2 for clarity and Figure 3 for layout

Reviewer 3 Report

Comments and Suggestions for Authors

Dear authors,

Thank you for submitting your work. Here are my comments for improvement of the paper:

-        Line 15: please write a full word and after that abbreviation, Sal-B, please do the same for all the abbreviations like OGD/R, Cx43, NLRP3, DisGenNET, etc.

-        Line 23, 24, 86, 111, 113;114, 122, 124, 128, 131; 155, 160, 167, 184, 192, 196, 218, 238, 305, 338, : Do not start a sentence with abbreviations, PPI; GO; KEGG, TTC, HE, TUNEL, Bcl2, Sal-B, Sirt1, ROS, MAPK, SIRT1, H9C2

-        Line 25, 31, 47, 68, 97, 137, : „in vivo“ and „in vitro“, „Salvia miltiorrhiza“ should be written in italics, please correct in the whole document

-        Line 62, 233, please put the reference at the end of the sentence

-        Lines 78 to 81 is not clear, please rewrite it and make it more clear, be precise and concise, the same is with 86 – 88; the same with 131- 139;

„Decide“ whether you should write SIRT1 or Sirt1 or sirt1, there is only one correct way to write it.

Part of the discussion should be put in the introduction part, like lines 176b- 177, 184 – 188, etc.

Correct line 229. there is no new method, only a better understanding of how Sal-B works and more information on this molecule to become a potential medicine.  

Line 272 correct space and define the abbreviation GO when mentioned first.

Improve the whole section of material and methods. It is very important to be very precise when explaining the material and methods in order to provide as much information as possible for other scientists so they can follow and repeat the procedure. It is not possible to do it from the current description of the methods. Write the origin of the cells used.

The general impression is that the work done is better than it is presented. The paper needs excessive editing not only in English grammar but much more in style. Sentences are not clear, there is no clear explanation of the terms used, it is full of abbreviations without previous explanations, and it lacks an in-depth approach to writing and explaining what was done and why it is important. In some parts, it is not clear if the work was done on cells, in animal models, or in silico. It is very hard to „catch“ the important connections since sentences are very short and unclear. Please put more effort into improvement. 

Comments on the Quality of English Language

 The paper needs excessive editing not only in English grammar but much more in style. Sentences are not clear, there is no clear explanation of the terms used, it is full of abbreviations without previous explanations, and it lacks an in-depth approach to writing and explaining what was done and why it is important. In some parts, it is not clear if the work was done on cells, in animal models, or in silico. It is very hard to „catch“ the important connections since sentences are very short and unclear. Please put more effort into improvement.  

Author Response

Dear editor and Reviewer,

Thank you for your careful reading and thoughtful comments on our previous manuscript. Those comments are very valuable and helpful to revise and improve our manuscript for us.

We have modified and polished the format and writing problems of the article, and marked it in red in the word document.

Round 2

Reviewer 1 Report

Comments and Suggestions for Authors

The revised form of the MS is promoted to interest for the readers including me. However, I think that a figure or cartoon that show the cell signaling pathway mediated by Salvianolic Acid B should be given in the summary section.

Author Response

Dear editors and reviewers: Thank you for your careful reading and thoughtful comments on our previous manuscript. We have revised the manuscript and added a graphical abstract.
